# Forecasting the Number of Road Accidents in Polish Provinces Using Trend Models

**Piotr Gorzelańczyk** 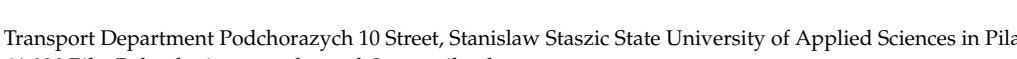

Transport Department Podchorazych 10 Street, Stanislaw Staszic State University of Applied Sciences in Pila, 64-920 Pila, Poland; piotr.gorzelanczyk@puss.pila.pl

**Abstract:** Many people die on the streets every year. The value is declining year by year, but there are still plenty of them. Although the COVID-19 pandemic reduced the number of traffic accidents, it is still very high. For this reason, in order to do everything possible to minimize the number of road accidents, it is important to know the federal states with the most road accidents and what the accident forecast is for the next few years. The purpose of this article is to predict the number of road accidents by state in Poland. The survey was divided into two parts. The first is an analysis of the annual data of police statistics on the number of road accidents in Poland for the period 2000–2021, upon the prediction of the number of traffic accidents from 2022 to 2031 was decided. The second part of the study looked at monthly data from 2000 to 2021. Again, the forecasts analyzed were determined for the period from January 2022 to December 2023. The results of this study indicate that a decrease in the number of accidents is also expected in the coming years, which becomes especially clear when analyzing the annual data. It is worth noting that the prevailing COVID-19 pandemic has distorted the results obtained. The study was performed in MS Excel using the selected propensity model.

**Keywords:** traffic accident; forecasting; trend models; province

## 1. Introduction

A traffic accident is an event that causes damage to property, as well as injury or death to traffic participants. According to the World Health Organization (WHO), about 1.3 million people die each year in road accidents. Those accidents account for about 3% of GDP in most countries around the world. Automobile accidents are the leading cause of death for minors and young people between the ages of 5 and 29 [1]. The United Nations General Assembly has set an ambitious goal of halving the number of traffic accident victims by 2030 [2,3].

Road accident severity is an attribute used to determine the severity of traffic collisions. Predicting accident severity is important for relevant authorities to develop road safety policies to prevent accidents and reduce injuries, fatalities and property damage [4–8]. Identifying key factors affecting accident severity is a prerequisite for taking measures to eliminate and reduce accident severity [9]. Yang et al. [10] proposed a DNN (deep neutral network) multicarbon structure for predicting various degrees of injury, death and severity of property loss. This allows for a comprehensive and accurate analysis of the severity of traffic accidents.

There are several sources of accident data. They are usually collected and evaluated by government agencies through responsible leadership agencies. Data are collected from police reports, insurance databases or hospital records. In some cases, traffic accident information is processed on a large scale for the transportation sector [11].

Intelligent transportation systems are currently the most important source of data for analyzing and predicting traffic accidents. These data can be processed using the vehicle's GPS device [12]. In addition, vehicle information can be detected using a roadside vehicle detection system, which can continuously record vehicle information (speed, traffic volume,

vehicle type, etc.) [13]. License plate recognition systems can also collect large amounts of traffic data during surveillance [14]. Another source of traffic and accident information is social media, but its accuracy can be insufficient due to the incompetence of reporters [15].

In this article, we chose a trend model to predict the number of traffic accidents by state. Exponential smoothing and neural networks for predicting the number of traffic accidents have been used by authors in other studies [16,17].

## 2. Literature Review

For accident data to be relevant, multiple data sources must be handled properly. Combining data from many sources, by merging various road accident data, increases the accuracy of analytical results [18].

A statistical study to estimate the severity of road accidents and clarify the relationship between the accidents and road users was conducted by Vilaca et al. [19]. The result of this research was suggestions for improving traffic safety standards and adopting other traffic safety measures.

Buck et al. [20] conducted a statistical study of traffic safety in selected regions of Poland based on the speed at which the number of road accidents and their causes were determined. The study used multivariate statistical analysis to examine the safety aspects of accident perpetrators.

The choice of the source of accident data for analysis depends on the type of traffic problem under consideration. Combining statistical models with other natural driving data or other data acquired by intelligent transportation systems can improve the accuracy of accident prediction and contribute to the elimination of accidents [21].

Various methods for predicting the number of accidents can be found in the literature. Time series methods are most commonly used to predict the number of traffic accidents [22,23]. The disadvantages are the inability to assess the quality of the prediction based on outdated forecasts and the frequent autocorrelation of the residual components of the accident rates [24]. Prochazka et al. [25] used multiple seasonality models for forecasting, and Sunny et al. [26] used Holt–Winters exponential smoothing. Its limitations include the inability to introduce exogenous variables into the model [27,28].

The vector autoregression model has also been used to predict the number of traffic accidents. The disadvantage is that it requires a large number of variable observations to properly estimate the parameters [29]. To analyze the number of deaths [30] and the Al-Madani curve-fitting regression model [31], they require only a simple linear relationship [32] and an order of autoregression (assuming the series is already stationary) [33].

Biswas et al. [34] used random forest regression to predict the number of traffic accidents. In this case, the data contain clusters of correlated features with similar validity to the original data, with small clusters favored over large clusters [35] and method instability and spike prediction [36]. Chudy-Laskowska and Pisula [37] applied autoregressive quadratic trend models, one-dimensional cyclical trend models and exponentially adjusted models to the forecasting problem at hand. Moving average models can also be used to predict the problem at hand. Its drawbacks are poor prediction accuracy, the resulting loss of data, and failure to account for trends and seasonal effects [38]. Prochozka and Camej [39] used the GARMA method. This method imposes some restrictions on the parameter space to ensure the stationarity of the process. ARMA models for steady-state processes and ARIMA or SARIMA models for transient processes are often used for forecasting [25,40,41]. Although these models lead to great flexibility of the models in question, they are also a disadvantage, as identifying good models requires more experience from the researcher than, for example, regression analysis [42]. Another disadvantage is the linearity of ARIMA models [43].

Chudy-Laskowska and Pisula used the ANOVA method in a study [44] to predict the number of traffic accidents. The disadvantage of this method is additional assumptions, especially the assumption of sphericity, the violation of which can lead to erroneous conclusions [45]. Neural network models are also used to predict the number of traffic accidents.

The disadvantages of SSNs are the need for experience in this area [44,46] and the dependence of the final solution on the initial conditions of the network and the lack of interpretability in the traditional way. As a black box, the model provides results as they come in, without any analytical knowledge [47].

A new method of prediction is the use of the Hadoop model by Kumar et al. [48]. The disadvantage of this method is that it does not support small data files [49]. Karlaftis and Vlahogianni [41] used the GARCH model for prediction. The disadvantage of this method is its complex formalism and complex model [50,51]. On the other hand, McIlroy and his team used the ADF test [52], which suffers from low power due to autocorrelation of random components [53]. Authors in other publications [54,55] have also used data mining techniques for prediction, but those usually suffer from a large number of general explanations [56]. Sevego et al. also found combinations of different models [57]. Parametric models have also been proposed in Bloomfield's work [58].

Given the above literature review, a quick and simple method for determining the forecast of the number of traffic accidents offered by MS Excel, such as trend models, was used for the study. Despite many studies using trend models, they have not been used to forecast the number of traffic accidents in Poland. For this reason, the author addressed the subject under discussion. Despite some limitations—it does not take into account the influence of seasonality in road accidents—it can be used to forecast the number of road accidents.

## 3. Materials and Methods

More than 38 million people live in Poland (Figure 1). It occupies an area of 312,705 km$^2$ and is divided into 16 provinces (Table 1, Figure 2). In the states studied, the average reduction in traffic accidents between 2001 and 2021 was more than 56%. Strongest in Kuyavsko–Pomorskie (70%) and Podraskiye (69%), lowest in Rubskiye (32%). The number of automobile accidents depends on the number of residents living in a particular state [59,60]. The number of accidents in Poland is still very high compared to the rest of the European Union [61]. Therefore, every effort should be made to reduce this value and identify the states with the most traffic accidents (Figures 3–5). Based on Figures 1 and 3, it can be concluded that the highest number of traffic accidents is in the Mazowieckie Province and the highest number of accidents per 10,000 inhabitants is in the Lodz Province.

In addition, since this study only used historical data on the number of traffic accidents and did not consider other factors, there is a relationship between the data used, and for this reason the method discussed can be used.

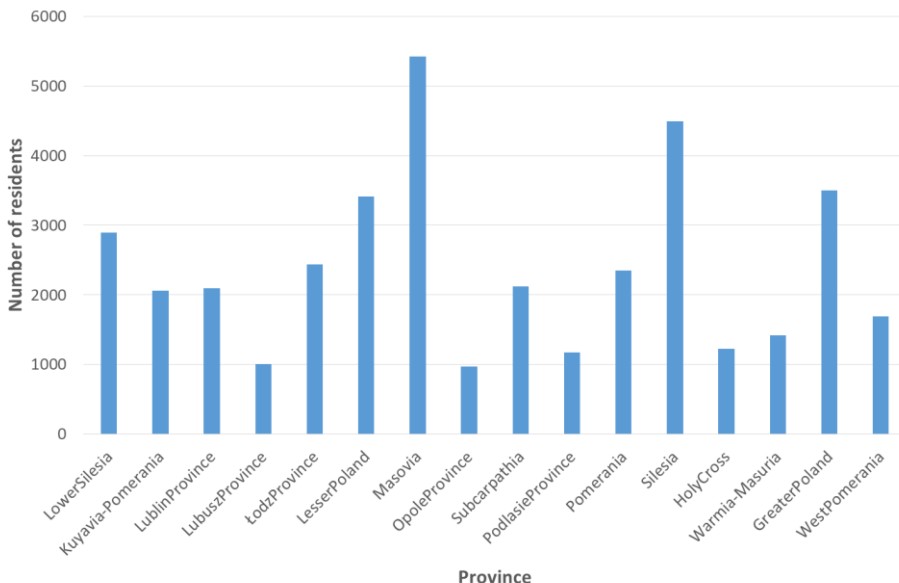

**Figure 1.** Population of Poland from 2001 to 2020 (thousand people) [62].

**Table 1.** Area, population by province in Poland in 2020 [62].

| Province | Area | Population | |
|---|---|---|---|
| | in km² | Total | Persons/km² |
| POLAND | 312,705 | 38,265,013 | 122 |
| Lower Silesia | 19,947 | 2,891,321 | 145 |
| Kuyavia–Pomerania | 17,971 | 2,061,942 | 115 |
| Lublin Province | 25,123 | 2,095,258 | 83 |
| Lubusz Province | 13,988 | 1,007,145 | 72 |
| Łodz Province | 18,219 | 2,437,970 | 134 |
| Lesser Poland | 15,183 | 3,410,441 | 225 |
| Masovia | 35,559 | 5,425,028 | 153 |
| Opole Province | 9412 | 976,774 | 104 |
| Subcarpathia | 17,846 | 2,121,229 | 119 |
| Podlasie Province | 20,187 | 1,173,286 | 58 |
| Pomerania | 18,323 | 2,346,671 | 128 |
| Silesia | 12,333 | 4,492,330 | 364 |
| Holy Cross | 11,710 | 1,224,626 | 105 |
| Warmia–Masuria | 24,173 | 1,416,495 | 59 |
| Greater Poland | 29,826 | 3,496,450 | 117 |
| West Pomerania | 22,905 | 1,688,047 | 74 |

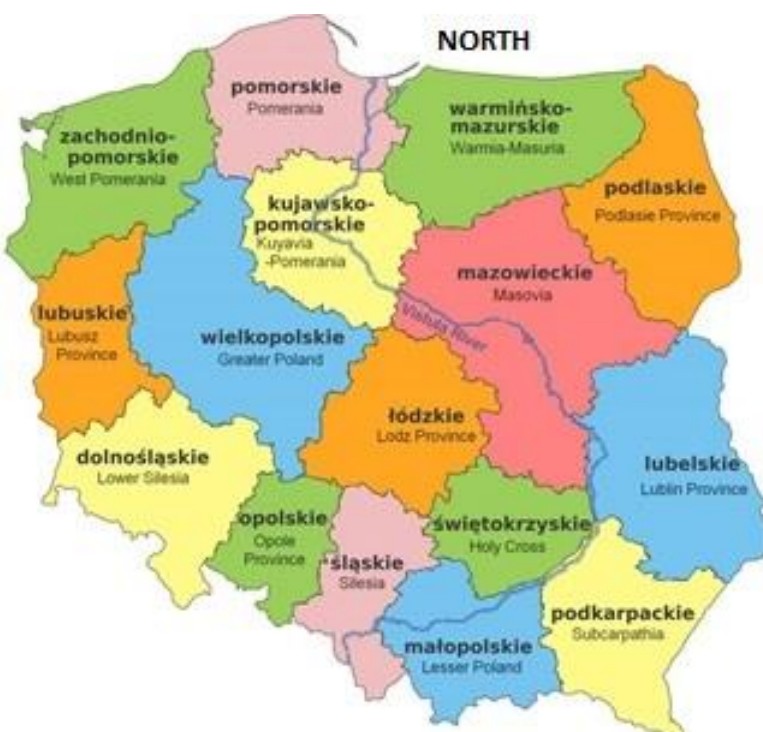

**Figure 2.** Location of provinces in Poland [62]. Scale: 1:1,000,000.

Considering Figures 3 and 4, it can be seen that the number of accidents is decreasing, and accidents are seasonal. The highest number of accidents occurs in the summer months and the lowest in the winter months. In addition, we can see that the beginning of the COVID-19 pandemic, 2020, disrupted the number of traffic accidents, when there was a huge decrease. Between 2001 and 2021, the largest decrease in the number of traffic accidents, more than 300%, is seen in the following provinces: kujawsko–pomorskie, lubelskie, podlaskie. On the other hand, the smallest decrease, during the analyzed period, was seen in Lubuskie province and amounted to less than 50%.

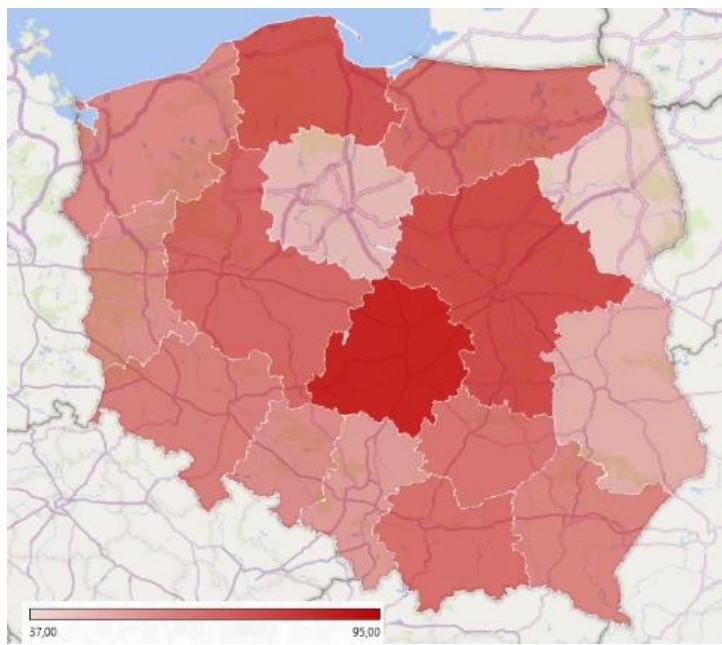

**Figure 3.** Accident number per 100,000 population in 2020 [62].

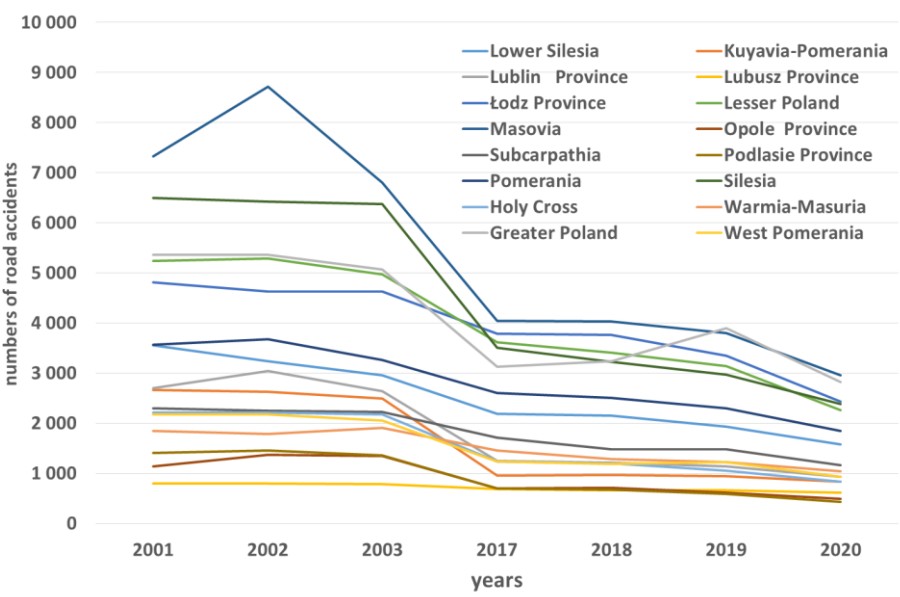

**Figure 4.** Number of road accidents in Poland by province from 2001 to 2021 [60].

The purpose of this article is to forecast the number of road accidents in Poland in each province. Statistical data of the Police from 2007 to 2021 were used as input data. The following trend models available in Excel software were used to forecast the number of road accidents in each province:

- Exponential;
- Linear;
- Logarithmic;
- 2nd degree polynomial;
- 3rd degree polynomial;
- Polynomial of the 4th degree;
- Polynomial of the 5th degree;
- Polynomial of the 6th degree;
- Potentiometric.

Then, for the obtained forecasts, the errors of expired forecasts were determined based on Equations (1)–(5):

- *ME*—mean error

$$ME = \frac{1}{n}\sum_{i=1}^{n}\left(Y_i - Y_p\right) \tag{1}$$

- *MAE*—mean average error

$$MAE = \frac{1}{n}\sum_{i=1}^{n}\left|Y_i - Y_p\right| \tag{2}$$

- *MPE*—mean percentage error

$$MPE = \frac{1}{n}\sum_{i=1}^{n}\frac{Y_i - Y_p}{Y_i} \tag{3}$$

- *MAPE*—mean absolute percentage error

$$MAPE = \frac{1}{n}\sum_{i=1}^{n}\frac{\left|Y_i - Y_p\right|}{Y_i} \tag{4}$$

- *MSE*—mean square error

$$MSE = \frac{1}{n}\sum_{i=1}^{n}\left(Y_i - Y_p\right)^2 \tag{5}$$

where

$n$—the length of the forecast horizon,
$Y$—observed value of road accidents,
$Y_p$—forecasted value of road accidents.

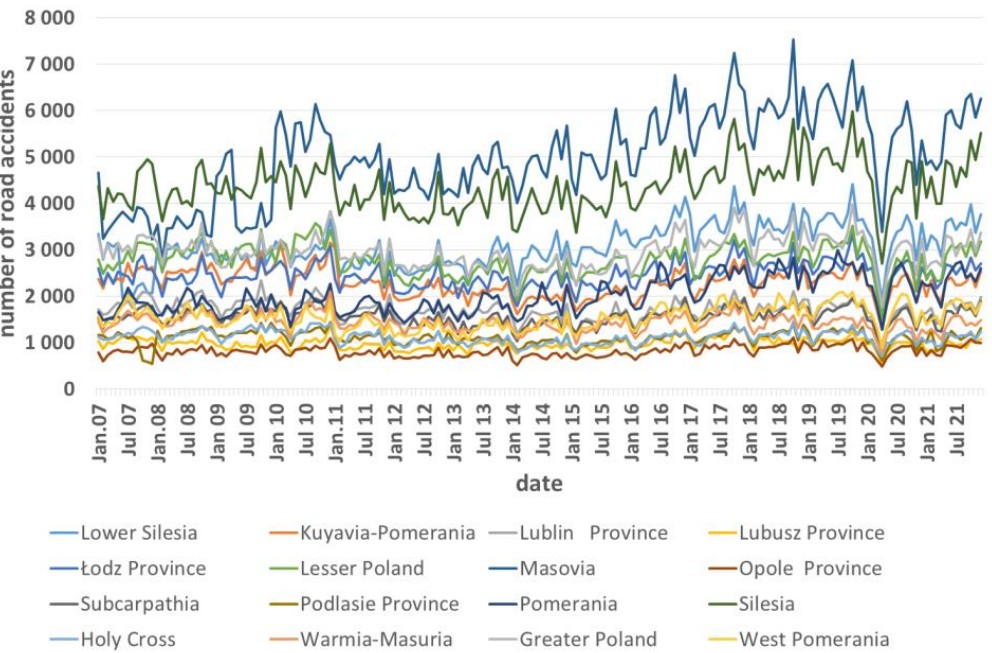

**Figure 5.** Number of road accidents in Poland by province from 2007 to 2021 [60].

## 4. Results

For the analyzed trend models, in the first step, the formulas for the statistical data analyzed on an annual and monthly basis for each federal state were determined using Excel software. For annual (2001 to 2021) and monthly (January 2007 to December 2021) data, serial graphs were drawn, and trend lines and R-squared values were determined. As can be seen, the R-squared coefficient, a measure of the quality of the model fit, is mostly good or fair for annual data, and mostly poor or satisfactory for monthly data, regardless of the model used. This is mainly due to the seasonality of traffic accident numbers in each state, which is not well accounted for by the methods used. Tables 2 and 3 and Appendix A shows trend model formulas with annual and monthly data for each state analyzed.

**Table 2.** Best trend models for annual data.

| Province | Model | Annual Data |
|---|---|---|
| Lower Silesia | Exponential | $y = 3660.6e^{-0.032x}$ |
| | | $R^2 = 0.7993$ |
| Kuyavia–Pomerania | Exponential | $y = 2968.8e^{-0.066x}$ |
| | | $R^2 = 0.9776$ |
| Lublin Province | Polynomial of 2nd degree | $y = 1.1254x^2 - 128.19x + 3070.1$ |
| | | $R^2 = 0.9649$ |
| Lubusz Province | Exponential | $y = 937.95e^{-0.019x}$ |
| | | $R^2 = 0.6083$ |
| Łodz Province | Exponential | $y = 5462.8e^{-0.027x}$ |
| | | $R^2 = 0.6749$ |
| Lesser Poland | Exponential | $y = 5751.5e^{-0.034x}$ |
| | | $R^2 = 0.7969$ |
| Masovia | Logarithmic | $y = -1744\ln(x) + 8908.6$ |
| | | $R^2 = 0.8619$ |
| Opole Province | Exponential | $y = 1384.8e^{-0.044x}$ |
| | | $R^2 = 0.9257$ |
| Subcarpathia | Exponential | $y = 2661.8e^{-0.032x}$ |
| | | $R^2 = 0.849$ |
| Podlasie Province | Exponential | $y = 1605.6e^{-0.057x}$ |
| | | $R^2 = 0.9468$ |
| Pomerania | Exponential | $y = 3663.6e^{-0.026x}$ |
| | | $R^2 = 0.7854$ |
| Silesian | Polynomial of 2nd degree | $y = -6.7177x^2 - 86.325x + 6865.4$ |
| | | $R^2 = 0.9765$ |
| Warmia–Masuria | Exponential | $y = 2545.8e^{-0.047x}$ |
| | | $R^2 = 0.9269$ |
| Holy Cross | Polynomial of 2nd degree | $y = -0.2957x^2 - 63.544x + 2323$ |
| | | $R^2 = 0.9656$ |
| Greater Poland | Potentiometric | $y = 6695.8x^{-0.301}$ |
| | | $R^2 = 0.6259$ |
| West Pomerania | Exponential | $y = 2381.2e^{-0.04x}$ |
| | | $R^2 = 0.939$ |

**Table 3.** Best trend models for monthly data.

| Province | Model | Monthly Data |
|---|---|---|
| **Lower Silesia** | Polynomial of 2nd degree | $y = 8\text{E-}06x^2 - 0.4824x + 9334.2$ |
| | | $R^2 = 0.3703$ |
| Kuyavia–Pomerania | Linear | $y = -0.0491x + 4343.8$ |
| | | $R^2 = 0.0651$ |
| Lublin Province | Linear | $y = -0.0316x + 3035.6$ |
| | | $R^2 = 0.0557$ |
| Lubusz Province | Exponential | $y = 900.21e^{2\text{E-}06x}$ |
| | | $R^2 = 0.0005$ |
| Łodz Province | Exponential | $y = 1286.6e^{2\text{E-}05x}$ |
| | | $R^2 = 0.0459$ |
| Lesser Poland | Linear | $y = -0.0187x + 3570.8$ |
| | | $R^2 = 0.0077$ |
| Masovia | Exponential | $y = 121.05e^{9\text{E-}05x}$ |
| | | $R^2 = 0.4903$ |
| Opole Province | Exponential | $y = 313.34e^{2\text{E-}05x}$ |
| | | $R^2 = 0.0597$ |
| Subcarpathia | Linear | $y = -0.0013x + 1628.5$ |
| | | $R^2 = 0.0001$ |
| Podlasie Province | Exponential | $y = 1308.3e^{-4\text{E-}06x}$ |
| | | $R^2 = 0.0019$ |
| Pomerania | Exponential | $y = 84.959e^{8\text{E-}05x}$ |
| | | $R^2 = 0.417$ |
| Silesian | Exponential | $y = 1795.9e^{2\text{E-}05x}$ |
| | | $R^2 = 0.0739$ |
| Holy Cross | Linear | $y = -0.0142x + 1713.4$ |
| | | $R^2 = 0.0243$ |
| Warmia-Masuria | Linear | $y = -0.0281x + 2606.2$ |
| | | $R^2 = 0.0597$ |
| Greater Poland | Exponential | $y = 2269.7e^{6\text{E-}06x}$ |
| | | $R^2 = 0.0067$ |
| West Pomerania | Exponential | $y = 389.87e^{3\text{E-}05x}$ |
| | | $R^2 = 0.0946$ |

We then used the data in Tables 2 and 3 and Appendix A to determine the number of traffic accident probabilities. For yearly data, this is the period from 2022 to 2031, and for monthly data, it is from January 2022 to December 2023. In this case, the prediction is based on data from police statistics. Prediction results using this method depend on the choice of model and its tuning.

A trend model with the lowest mean percentage error and mean absolute percentage error was chosen to predict the number of traffic accidents in the analyzed federal states. Based on this, we found that for annual statistical data, the best model is often the exponential model, which has the smallest analyzed error. In addition, polynomial, logarithmic and power

models were also used for annual data. In this case, the average MAPE error was 0.52%. On the other hand, for monthly data, linear and exponential models produce the smallest MAPE errors depending on the states studied. However, the average for this error was 65%. This is a very large value and suggests that the propensity model should not be used to predict the number of traffic accidents. Tables 4 and 5 summarize the errors between the lowest modeled annual and monthly data. Using these models, the predicted number of accidents for the next year was determined on a monthly and yearly basis (Figures 6 and 7). Based on Figures 4 and 5, the number of traffic accidents is expected to decline further in the next few years. Please note that the COVID-19 pandemic has changed our forecast significantly. As can be seen from Figure 6, the trend model does not account for the seasonality that occurs in traffic accidents and should not be used in the case under consideration.

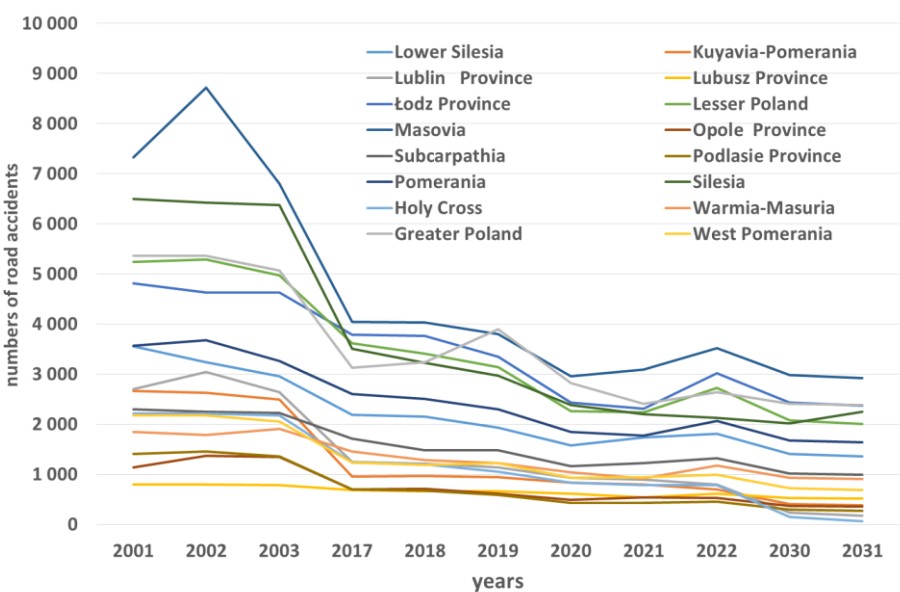

**Figure 6.** Forecasting annual number of road accidents for 2022–2031.

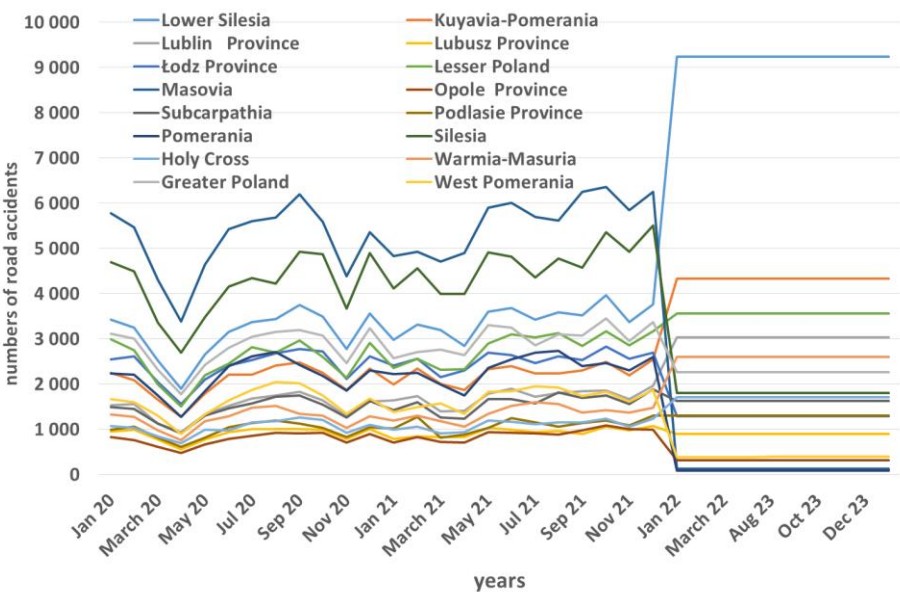

**Figure 7.** Forecasting monthly number of road accidents for 2022–2023.

**Table 4.** Summary of errors for annual data.

| Province/Model | Model | ME | MPE | The Sum of the Squares | MSE | MAPE (%) | MAE (%) |
|---|---|---|---|---|---|---|---|
| Lower Silesia | Exponential | 14.26 | 184.47 | 1,305,179.18 | 62,151.39 | 0.26 | 7.22 |
| Kuyavia–Pomerania | Exponential | 10.95 | 70.32 | 149,986.76 | 7142.23 | 0.35 | 5.00 |
| Lublin Province | Polynomial of 2nd degree | 0.07 | 93.93 | 301,100.51 | 14,338.12 | 0.32 | 5.09 |
| Lubusz Province | Exponential | 2.76 | 55.03 | 119,113.40 | 5672.07 | 0.47 | 6.93 |
| Łodz Province | Exponential | 6.86 | 317.71 | 3,302,784.34 | 157,275.44 | 1.01 | 8.72 |
| Lesser Poland | Exponential | 26.74 | 290.75 | 2,392,974.36 | 11,3951.16 | 0.14 | 8.17 |
| Masovia | Logarithmic | 176.78 | 497.81 | 10,533,523.06 | 501,596.34 | 1.77 | 8.75 |
| Opole Province | Exponential | 4.25 | 54.32 | 107,325.17 | 5110.72 | 0.02 | 5.98 |
| Subcarpathia | Exponential | 11.57 | 120.66 | 495,205.41 | 23,581.21 | 0.11 | 6.41 |
| Podlasie Province | Exponential | 2.39 | 52.82 | 81,724.62 | 3891.65 | 0.20 | 6.66 |
| Pomerania | Exponential | 12.34 | 168.89 | 853,023.82 | 40,620.18 | 0.14 | 6.61 |
| Silesia | Polynomial of 3rd degree | 0.02 | 144.78 | 755,411.76 | 35,971.99 | 0.17 | 3.24 |
| Holy Cross | Polynomial of 2nd degree | 0.01 | 72.91 | 134,675.61 | 6413.12 | 0.41 | 5.26 |
| Warmia–Masuria | Exponential | 4.82 | 150.77 | 765,403.02 | 36,447.76 | 1.00 | 9.60 |
| Greater Poland | Exponential | 48.45 | 563.32 | 9,570,955.89 | 455,759.80 | 1.81 | 16.12 |
| West Pomerania | Exponential | 1.99 | 75.89 | 166,539.04 | 7930.43 | 0.16 | 5.12 |

**Table 5.** Summary of errors for monthly data.

| Province/Model | Model | ME | MPE | The Sum of the Squares | MSE | MAPE (%) | MAE (%) |
|---|---|---|---|---|---|---|---|
| Lower Silesia | Polynomial of 2nd degree | 6201.42 | 6201.42 | 6,965,812,277.61 | 38,698,957.10 | 208.40 | 208.40 |
| Kuyavia–Pomerania | Linear | 2046.27 | 2046.27 | 770,270,675.15 | 4,279,281.53 | 92.75 | 92.75 |
| Lublin Province | Linear | 1316.42 | 1316.42 | 319,951,089.84 | 1,777,506.05 | 79.68 | 79.68 |
| Lubusz Province | Exponential | 75.75 | 111.95 | 3,708,472.78 | 20,602.63 | 6.29 | 10.97 |
| Łodz Province | Exponential | 1175.17 | 1175.17 | 262,350,703.45 | 1,457,503.91 | 47.01 | 47.01 |
| Lesser Poland | Linear | 778.52 | 778.59 | 129,389,638.72 | 718,831.33 | 29.95 | 29.95 |
| Masovia | Exponential | 4894.72 | 4894.72 | 4,479,240,358.37 | 24,884,668.66 | 97.47 | 97.47 |
| Opole Province | Exponential | 502.58 | 502.58 | 47,967,460.50 | 266,485.89 | 60.72 | 60.72 |
| Subcarpathia | Linear | 56.11 | 156.75 | 7,374,357.20 | 40,968.65 | 5.31 | 10.83 |
| Podlasie Province | Exponential | 198.30 | 206.68 | 11,259,059.55 | 62,550.33 | 20.66 | 21.26 |
| Pomerania | Exponential | 1937.33 | 1937.33 | 701,204,745.03 | 3,895,581.92 | 95.63 | 95.63 |
| Silesia | Exponential | 2575.68 | 2575.68 | 1,245,839,964.63 | 6,921,333.14 | 58.25 | 58.25 |
| Holy Cross | Linear | 591.69 | 591.69 | 66,739,299.40 | 370,773.89 | 55.47 | 55.47 |
| Warmia–Masuria | Linear | 1171.74 | 1171.74 | 253,066,000.67 | 1,405,922.23 | 85.00 | 85.00 |
| Greater Poland | Exponential | 699.20 | 707.79 | 110,227,747.58 | 612,376.38 | 22.44 | 22.90 |
| West Pomerania | Exponential | 1157.77 | 1157.77 | 252,984,847.29 | 1,405,471.37 | 74.04 | 74.04 |

## 5. Conclusions

Forecasts of the number of accidents in Poland for individual provinces were determined by selected trend models using Excel. The results show that we can still expect a decrease in the number of traffic accidents in the coming years. It should be noted that the COVID-19 pandemic has distorted the results obtained, and if it continues and traffic restrictions are introduced, the proposed model may not be adequate. The value of the average error of 0.52% for annual data can testify to the choice of an effective forecasting

method. As we can see, trend models fail for forecasting the monthly number of traffic accidents in which there is seasonality. On the other hand, for annual data, the results are at a high level. The advantage of trend models is their speed of determining the forecast.

The forecast number of traffic accidents obtained in the article can be used, in the future, to formulate further measures to minimize the number of accidents in the analyzed country. These measures may include, for example, the introduction of higher fines for traffic offenses on Polish roads from 1 January 2022.

In his further research, the author plans to take into account more factors affecting the accident rate in Poland and use other methods to forecast the number of road accidents. These may include, but are not limited to traffic volume, day of the week or the age of the perpetrator of the accident and also conduct research using other methods for forecasting the number of road accidents in Poland, such as neural networks and exponential smoothing.

**Funding:** The article was financed by the university's own funds.

**Institutional Review Board Statement:** Not applicable.

**Informed Consent Statement:** Not applicable.

**Data Availability Statement:** The article was written on the basis of public data available on the pages of the Police Department.

**Conflicts of Interest:** The author declares no conflict of interest.

## Appendix A

**Table A1.** Trend models for the Lower Silesian province.

| Data/Model | Annual Data | Monthly Data |
|---|---|---|
| Exponential | $y = 3660.6e^{-0.032x}$ | $y = 1\text{E-}08x^{2.4702}$ |
| | $R^2 = 0.7993$ | $R^2 = 0.3518$ |
| Linear | $y = -79.014x + 3506.4$ | $y = 0.1826x - 4546.7$ |
| | $R^2 = 0.8145$ | $R^2 = 0.3689$ |
| Logarithmic | $y = -539.9\ln(x) + 3804.1$ | $y = 7618.2\ln(x) - 77969$ |
| | $R^2 = 0.6604$ | $R^2 = 0.3679$ |
| Polynomial of 2nd degree | $y = -2.662x^2 - 20.451x + 3282$ | $y = 8\text{E-}06x^2 - 0.4824x + 9334.2$ |
| | $R^2 = 0.8415$ | $R^2 = 0.3703$ |
| Polynomial of 3rd degree | $y = -0.3402x^3 + 8.5652x^2 - 121.56x + 3488.5$ | $y = -3\text{E-}08x^3 + 0.0041x^2 - 171.09x + 2\text{E+}06$ |
| | $R^2 = 0.8537$ | $R^2 = 0.4156$ |
| Polynomial of 4th degree | $y = 0.0557x^4 - 2.789x^3 + 43.763x^2 - 303.3x + 3729.9$ | $y = -2\text{E-}11x^4 + 3\text{E-}06x^3 - 0.1807x^2 + 4971.6x - 5\text{E+}07$ |
| | $R^2 = 0.8625$ | $R^2 = 0.4408$ |
| Polynomial of 5th degree | $y = -0.0019x^5 + 0.1617x^4 - 4.8892x^3 + 61.752x^2 - 365.45x + 3790.3$ | $y = 2\text{E-}14x^5 - 3\text{E-}09x^4 + 0.0003x^3 - 12.199x^2 + 255831x - 2\text{E+}09$ |
| | $R^2 = 0.8628$ | $R^2 = 0.4826$ |
| Polynomial of 6th degree | $y = 0.002x^6 - 0.1329x^5 + 3.4719x^4 - 44.893x^3 + 296.45x^2 - 961.09x + 4248$ | $y = 2\text{E-}17x^6 - 4\text{E-}12x^5 + 4\text{E-}07x^4 - 0.0233x^3 + 727.3x^2 - 1\text{E+}07x + 8\text{E+}10$ |
| | $R^2 = 0.8703$ | $R^2 = 0.5586$ |
| Potentiometric | $y = 4062.9x^{-0.21}$ | $y = 1\text{E-}08x^{2.4702}$ |
| | $R^2 = 0.6058$ | $R^2 = 0.3518$ |

**Table A2.** Trend models for the Kuyavian–Pomeranian province.

| Data/Model | Annual Data | Monthly Data |
|---|---|---|
| Exponential | $y = 2968.8e^{-0.066x}$ | $y = 5563.1e^{-2E-05x}$ |
| | $R^2 = 0.9776$ | $R^2 = 0.0624$ |
| Linear | $y = -100.67x + 2672.2$ | $y = -0.0491x + 4343.8$ |
| | $R^2 = 0.9481$ | $R^2 = 0.0651$ |
| Logarithmic | $y = -749.4\ln(x) + 3184.3$ | $y = -2093\ln(x) + 24565$ |
| | $R^2 = 0.9122$ | $R^2 = 0.068$ |
| Polynomial of 2nd degree | $y = 3.7309x^2 - 182.75x + 2986.9$ | $y = 4E\text{-}05x^2 - 3.7996x + 82631$ |
| | $R^2 = 0.9861$ | $R^2 = 0.174$ |
| Polynomial of 3rd degree | $y = 0.2451x^3 - 4.357x^2 - 109.91x + 2838.1$ | $y = -2E\text{-}09x^3 + 0.0003x^2 - 14.473x + 231109$ |
| | $R^2 = 0.9906$ | $R^2 = 0.1745$ |
| Polynomial of 4th degree | $y = -0.0404x^4 + 2.0229x^3 - 29.91x^2 + 22.025x + 2662.8$ | $y = -3E\text{-}11x^4 + 5E\text{-}06x^3 - 0.313x^2 + 8706.8x - 9E+07$ |
| | $R^2 = 0.994$ | $R^2 = 0.3524$ |
| Polynomial of 5th degree | $y = -0.0044x^5 + 0.1989x^4 - 2.7169x^3 + 10.689x^2 - 118.23x + 2799.1$ | $y = 1E\text{-}14x^5 - 2E\text{-}09x^4 + 0.0002x^3 - 7.3524x^2 + 155643x - 1E+09$ |
| | $R^2 = 0.995$ | $R^2 = 0.3874$ |
| Polynomial of 6th degree | $y = -0.0001x^6 + 0.0026x^5 + 0.0235x^4 - 0.5973x^3 - 1.7467x^2 - 86.666x + 2774.9$ | $y = 1E\text{-}17x^6 - 3E\text{-}12x^5 + 3E\text{-}07x^4 - 0.0159x^3 + 494.62x^2 - 8E+06x + 6E+10$ |
| | $R^2 = 0.995$ | $R^2 = 0.4732$ |
| Potentiometric | $y = 3938.4x^{-0.464}$ | $y = 4E+07x^{-0.914}$ |
| | $R^2 = 0.8524$ | $R^2 = 0.0652$ |

**Table A3.** Trend models for Lublin province.

| Data/Model | Annual Data | Monthly Data |
|---|---|---|
| Exponential | $y = 3305.9e^{-0.059x}$ | $y = 3789.7e^{-2E-05x}$ |
| | $R^2 = 0.9677$ | $R^2 = 0.0549$ |
| Linear | $y = -103.43x + 2975.2$ | $y = -0.0316x + 3035.6$ |
| | $R^2 = 0.9615$ | $R^2 = 0.0557$ |
| Logarithmic | $y = -733.2\ln(x) + 3421.7$ | $y = -1340\ln(x) + 15972$ |
| | $R^2 = 0.8388$ | $R^2 = 0.0576$ |
| Polynomial of 2nd degree | $y = 1.1254x^2 - 128.19x + 3070.1$ | $y = 2E\text{-}05x^2 - 1.8755x + 41525$ |
| | $R^2 = 0.9649$ | $R^2 = 0.1102$ |
| Polynomial of 3rd degree | $y = 0.2624x^3 - 7.5334x^2 - 50.211x + 2910.7$ | $y = 7E\text{-}10x^3 - 7E\text{-}05x^2 + 1.9618x - 11858$ |
| | $R^2 = 0.9699$ | $R^2 = 0.1103$ |
| Polynomial of 4th degree | $y = -0.0316x^4 + 1.6547x^3 - 27.546x^2 + 53.118x + 2773.5$ | $y = -1E\text{-}11x^4 + 2E\text{-}06x^3 - 0.1304x^2 + 3628.8x - 4E+07$ |
| | $R^2 = 0.9718$ | $R^2 = 0.174$ |
| Polynomial of 5th degree | $y = -0.0058x^5 + 0.2879x^4 - 4.6744x^3 + 26.667x^2 - 134.16x + 2955.5$ | $y = 7E\text{-}15x^5 - 1E\text{-}09x^4 + 0.0001x^3 - 5.1059x^2 + 107484x - 9E+08$ |
| | $R^2 = 0.9736$ | $R^2 = 0.2102$ |
| Polynomial of 6th degree | $y = -0.0012x^6 + 0.0765x^5 - 1.7937x^4 + 20.481x^3 - 120.92x^2 + 240.4x + 2667.7$ | $y = 6E\text{-}18x^6 - 1E\text{-}12x^5 + 2E\text{-}07x^4 - 0.0086x^3 + 267.23x^2 - 4E+06x + 3E+10$ |
| | $R^2 = 0.9756$ | $R^2 = 0.2624$ |
| Potentiometric | $y = 4104.7x^{-0.402}$ | $y = 1E+07x^{-0.812}$ |
| | $R^2 = 0.7726$ | $R^2 = 0.0567$ |

**Table A4.** Trend models for Lubuskie province.

| Data/Model | Annual Data | Monthly Data |
|---|---|---|
| Exponential | $y = 937.95e^{-0.019x}$ | $y = 900.21e^{2E-06x}$ |
| | $R^2 = 0.6083$ | $R^2 = 0.0005$ |
| Linear | $y = -13.829x + 920.97$ | $y = 0.0027x + 862.02$ |
| | $R^2 = 0.5766$ | $R^2 = 0.0013$ |
| Logarithmic | $y = -73.48\ln(x) + 927.65$ | $y = 106.88\ln(x) - 161.11$ |
| | $R^2 = 0.2827$ | $R^2 = 0.0011$ |
| Polynomial of 2nd degree | $y = -1.6013x^2 + 21.401x + 785.93$ | $y = 7E\text{-}06x^2 - 0.5564x + 12532$ |
| | $R^2 = 0.8019$ | $R^2 = 0.0163$ |
| Polynomial of 3rd degree | $y = 0.1435x^3 - 6.3358x^2 + 64.039x + 698.81$ | $y = -1E\text{-}08x^3 + 0.0017x^2 - 69.977x + 978278$ |
| | $R^2 = 0.8521$ | $R^2 = 0.1304$ |
| Polynomial of 4th degree | $y = 0.0061x^4 - 0.1246x^3 - 2.4829x^2 + 44.145x + 725.24$ | $y = -5E\text{-}12x^4 + 8E\text{-}07x^3 - 0.0477x^2 + 1303.7x - 1E+07$ |
| | $R^2 = 0.8546$ | $R^2 = 0.1579$ |
| Polynomial of 5th degree | $y = -0.0067x^5 + 0.3758x^4 - 7.4473x^3 + 60.24x^2 - 172.54x + 935.78$ | $y = 4E\text{-}15x^5 - 9E\text{-}10x^4 + 8E\text{-}05x^3 - 3.3252x^2 + 69716x - 6E+08$ |
| | $R^2 = 0.9327$ | $R^2 = 0.2052$ |
| Polynomial of 6th degree | $y = -0.0003x^6 + 0.0147x^5 - 0.1649x^4 - 0.9136x^3 + 21.908x^2 - 75.253x + 861.02$ | $y = 4E\text{-}18x^6 - 1E\text{-}12x^5 + 1E\text{-}07x^4 - 0.0056x^3 + 174.52x^2 - 3E+06x + 2E+10$ |
| | $R^2 = 0.9373$ | $R^2 = 0.2721$ |
| Potentiometric | $y = 949.27x^{-0.102}$ | $y = 481.83x^{0.0656}$ |
| | $R^2 = 0.3059$ | $R^2 = 0.0004$ |

**Table A5.** Trend models for Łódź Province.

| Data/Model | Annual Data | Monthly Data |
|---|---|---|
| Exponential | $y = 5462.8e^{-0.027x}$ | $y = 1286.6e^{2E-05x}$ |
| | $R^2 = 0.6749$ | $R^2 = 0.0459$ |
| Linear | $y = -101x + 5231.5$ | $y = 0.0403x + 778.52$ |
| | $R^2 = 0.7447$ | $R^2 = 0.053$ |
| Logarithmic | $y = -624.7\ln(x) + 5470.4$ | $y = 1658.5\ln(x) - 15182$ |
| | $R^2 = 0.4947$ | $R^2 = 0.0515$ |
| Polynomial of 2nd degree | $y = -7.4751x^2 + 63.449x + 4601.1$ | $y = 2E\text{-}05x^2 - 2.0299x + 43991$ |
| | $R^2 = 0.8636$ | $R^2 = 0.0931$ |
| Polynomial of 3rd degree | $y = -0.5521x^3 + 10.745x^2 - 100.64x + 4936.3$ | $y = -1E\text{-}08x^3 + 0.0017x^2 - 73.911x + 1E+06$ |
| | $R^2 = 0.8816$ | $R^2 = 0.1168$ |
| Polynomial of 4th degree | $y = -0.1513x^4 + 6.1071x^3 - 84.97x^2 + 393.56x + 4279.9$ | $y = -2E\text{-}11x^4 + 3E\text{-}06x^3 - 0.1873x^2 + 5189.5x - 5E+07$ |
| | $R^2 = 0.9181$ | $R^2 = 0.195$ |
| Polynomial of 5th degree | $y = -0.0197x^5 + 0.9343x^4 - 15.398x^3 + 99.229x^2 - 242.77x + 4898.2$ | $y = 4E\text{-}15x^5 - 9E\text{-}10x^4 + 7E\text{-}05x^3 - 3.1414x^2 + 66850x - 6E+08$ |
| | $R^2 = 0.9344$ | $R^2 = 0.2024$ |
| Polynomial of 6th degree | $y = 0.0044x^6 - 0.3069x^5 + 8.1925x^4 - 103.11x^3 + 613.84x^2 - 1548.8x + 5901.8$ | $y = 1E\text{-}17x^6 - 3E\text{-}12x^5 + 3E\text{-}07x^4 - 0.0154x^3 + 479.91x^2 - 8E+06x + 6E+10$ |
| | $R^2 = 0.9546$ | $R^2 = 0.2982$ |
| Potentiometric | $y = 5778.5x^{-0.165}$ | $y = 2.9338x^{0.6322}$ |
| | $R^2 = 0.428$ | $R^2 = 0.0444$ |

**Table A6.** Trend models for the Lesser Poland province.

| Data/Model | Annual Data | Monthly Data |
|---|---|---|
| Exponential | $y = 5751.5e^{-0.034x}$ | $y = 3818.4e^{-8E-06x}$ |
| | $R^2 = 0.7969$ | $R^2 = 0.0093$ |
| Linear | $y = -125.74x + 5453.3$ | $y = -0.0187x + 3570.8$ |
| | $R^2 = 0.8736$ | $R^2 = 0.0077$ |
| Logarithmic | $y = -856.4\ln(x) + 5920.7$ | $y = -816\ln(x) + 11472$ |
| | $R^2 = 0.7035$ | $R^2 = 0.0085$ |
| Polynomial of 2nd degree | $y = -4.2552x^2 - 32.128x + 5094.5$ | $y = 4E\text{-}05x^2 - 3.0715x + 67295$ |
| | $R^2 = 0.9027$ | $R^2 = 0.0669$ |
| Polynomial of 3rd degree | $y = -0.9622x^3 + 27.499x^2 - 318.11x + 5678.7$ | $y = -4E\text{-}09x^3 + 0.0005x^2 - 24.164x + 360727$ |
| | $R^2 = 0.9441$ | $R^2 = 0.0683$ |
| Polynomial of 4th degree | $y = -0.0701x^4 + 2.1228x^3 - 16.844x^2 - 89.152x + 5374.6$ | $y = -3E\text{-}11x^4 + 5E\text{-}06x^3 - 0.3065x^2 + 8523.8x - 9E+07$ |
| | $R^2 = 0.9501$ | $R^2 = 0.2084$ |
| Polynomial of 5th degree | $y = -0.0057x^5 + 0.2409x^4 - 4.0384x^3 + 35.93x^2 - 271.46x + 5551.8$ | $y = 2E\text{-}14x^5 - 4E\text{-}09x^4 + 0.0003x^3 - 12.543x^2 + 263934x - 2E+09$ |
| | $R^2 = 0.9511$ | $R^2 = 0.2952$ |
| Polynomial of 6th degree | $y = 0.0014x^6 - 0.1005x^5 + 2.6393x^4 - 33.023x^3 + 205.98x^2 - 703.03x + 5883.4$ | $y = 1E\text{-}17x^6 - 3E\text{-}12x^5 + 3E\text{-}07x^4 - 0.0176x^3 + 547.22x^2 - 9E+06x + 6E+10$ |
| | $R^2 = 0.9527$ | $R^2 = 0.3826$ |
| Potentiometric | $y = 6386.4x^{-0.219}$ | $y = 96700x^{-0.334}$ |
| | $R^2 = 0.5906$ | $R^2 = 0.0101$ |

**Table A7.** Trend models for the Masovia province.

| Data/Model | Annual Data | Monthly Data |
|---|---|---|
| Exponential | $y = 8271.5e^{-0.048x}$ | $y = 121.05e^{9E-05x}$ |
| | $R^2 = 0.9283$ | $R^2 = 0.4903$ |
| Linear | $y = -244.75x + 7786.2$ | $y = 0.4231x - 12672$ |
| | $R^2 = 0.8992$ | $R^2 = 0.483$ |
| Logarithmic | $y = -1744\ln(x) + 8908.6$ | $y = 17743\ln(x) - 183769$ |
| | $R^2 = 0.8619$ | $R^2 = 0.4869$ |
| Polynomial of 2nd degree | $y = 4.749x^2 - 344.48x + 8151.9$ | $y = -8E\text{-}05x^2 + 6.7035x - 143764$ |
| | $R^2 = 0.9082$ | $R^2 = 0.5134$ |
| Polynomial of 3rd degree | $y = -0.1364x^3 + 9.0453x^2 - 381.46x + 8224.4$ | $y = -2E\text{-}08x^3 + 0.0026x^2 - 104.17x + 1E+06$ |
| | $R^2 = 0.9084$ | $R^2 = 0.5181$ |
| Polynomial of 4th degree | $y = 0.0347x^4 - 1.593x^3 + 29.046x^2 - 480.3x + 8350.7$ | $y = -4E\text{-}11x^4 + 6E\text{-}06x^3 - 0.3799x^2 + 10544x - 1E+08$ |
| | $R^2 = 0.9086$ | $R^2 = 0.5446$ |
| Polynomial of 5th degree | $y = -0.0841x^5 + 4.4507x^4 - 85.146x^3 + 713.52x^2 - 2747.9x + 10479$ | $y = 8E\text{-}15x^5 - 2E\text{-}09x^4 + 0.0001x^3 - 6.0186x^2 + 128241x - 1E+09$ |
| | $R^2 = 0.9493$ | $R^2 = 0.5468$ |
| Polynomial of 6th degree | $y = 0.0162x^6 - 1.1063x^5 + 29.128x^4 - 370.28x^3 + 2315.8x^2 - 6656.5x + 13390$ | $y = 4E\text{-}17x^6 - 1E\text{-}11x^5 + 1E\text{-}06x^4 - 0.0603x^3 + 1885.7x^2 - 3E+07x + 2E+11$ |
| | $R^2 = 0.984$ | $R^2 = 0.6682$ |
| Potentiometric | $y = 9945.1x^{-0.324}$ | $y = 3E\text{-}14x^{3.7203}$ |
| | $R^2 = 0.8059$ | $R^2 = 0.4954$ |

**Table A8.** Trend models for Opole province.

| Data/Model | Annual Data | Monthly Data |
|---|---|---|
| Exponential | $y = 1384.8e^{-0.044x}$ | $y = 313.34e^{2E-05x}$ |
| | $R^2 = 0.9257$ | $R^2 = 0.0597$ |
| Linear | $y = -37.996x + 1306.3$ | $y = 0.0201x - 25.195$ |
| | $R^2 = 0.909$ | $R^2 = 0.0728$ |
| Logarithmic | $y = -270\ln(x) + 1471.8$ | $y = 826.22\ln(x) - 7974.5$ |
| | $R^2 = 0.797$ | $R^2 = 0.0703$ |
| Polynomial of 2nd degree | $y = 0.5472x^2 - 50.034x + 1352.4$ | $y = 1E-05x^2 - 1.1854x + 25138$ |
| | $R^2 = 0.9145$ | $R^2 = 0.1476$ |
| Polynomial of 3rd degree | $y = 0.0029x^3 + 0.451x^2 - 49.168x + 1350.7$ | $y = -3E-09x^3 + 0.0004x^2 - 15.737x + 227570$ |
| | $R^2 = 0.9145$ | $R^2 = 0.153$ |
| Polynomial of 4th degree | $y = -0.0389x^4 + 1.7163x^3 - 24.176x^2 + 77.988x + 1181.8$ | $y = -8E-12x^4 + 1E-06x^3 - 0.0787x^2 + 2184.9x - 2E+07$ |
| | $R^2 = 0.9353$ | $R^2 = 0.2282$ |
| Polynomial of 5th degree | $y = 0.0044x^5 - 0.281x^4 + 6.5105x^3 - 65.24x^2 + 219.85x + 1043.9$ | $y = 4E-15x^5 - 8E-10x^4 + 7E-05x^3 - 2.8588x^2 + 60215x - 5E+08$ |
| | $R^2 = 0.9423$ | $R^2 = 0.2645$ |
| Polynomial of 6th degree | $y = -0.0006x^6 + 0.0425x^5 - 1.2443x^4 + 18.152x^3 - 133.54x^2 + 393.19x + 910.74$ | $y = 4E-18x^6 - 1E-12x^5 + 1E-07x^4 - 0.0064x^3 + 199.59x^2 - 3E+06x + 2E+10$ |
| | $R^2 = 0.9454$ | $R^2 = 0.3572$ |
| Potentiometric | $y = 1636x^{-0.3}$ | $y = 0.0415x^{0.9283}$ |
| | $R^2 = 0.7549$ | $R^2 = 0.0575$ |

**Table A9.** Trend models for the Subcarpathia province.

| Data/Model | Annual Data | Monthly Data |
|---|---|---|
| Exponential | $y = 2661.8e^{-0.032x}$ | $y = 1690.1e^{-2E-06x}$ |
| | $R^2 = 0.849$ | $R^2 = 0.0006$ |
| Linear | $y = -56.701x + 2542.6$ | $y = -0.0013x + 1628.5$ |
| | $R^2 = 0.8799$ | $R^2 = 0.0001$ |
| Logarithmic | $y = -359.2\ln(x) + 2695$ | $y = -74.05\ln(x) + 2360.1$ |
| | $R^2 = 0.613$ | $R^2 = 0.0002$ |
| Polynomial of 2nd degree | $y = -2.7746x^2 + 4.3389x + 2308.6$ | $y = 2E-05x^2 - 1.4932x + 32768$ |
| | $R^2 = 0.9413$ | $R^2 = 0.0423$ |
| Polynomial of 3rd degree | $y = 0.1297x^3 - 7.0544x^2 + 42.883x + 2229.9$ | $y = -2E-09x^3 + 0.0002x^2 - 9.9627x + 150593$ |
| | $R^2 = 0.945$ | $R^2 = 0.0429$ |
| Polynomial of 4th degree | $y = -0.022x^4 + 1.0973x^3 - 20.961x^2 + 114.69x + 2134.5$ | $y = -2E-11x^4 + 3E-06x^3 - 0.1615x^2 + 4490.4x - 5E+07$ |
| | $R^2 = 0.9479$ | $R^2 = 0.1587$ |
| Polynomial of 5th degree | $y = -0.0062x^5 + 0.32x^4 - 5.676x^3 + 37.055x^2 - 85.735x + 2329.3$ | $y = 8E-15x^5 - 2E-09x^4 + 0.0001x^3 - 5.7557x^2 + 121260x - 1E+09$ |
| | $R^2 = 0.954$ | $R^2 = 0.2129$ |
| Polynomial of 6th degree | $y = 0.0012x^6 - 0.0856x^5 + 2.3262x^4 - 29.922x^3 + 179.3x^2 - 446.75x + 2606.7$ | $y = 8E-18x^6 - 2E-12x^5 + 2E-07x^4 - 0.0116x^3 + 360.56x^2 - 6E+06x + 4E+10$ |
| | $R^2 = 0.9598$ | $R^2 = 0.3245$ |
| Potentiometric | $y = 2875.5x^{-0.197}$ | $y = 4146.2x^{-0.092}$ |
| | $R^2 = 0.5705$ | $R^2 = 0.0007$ |

**Table A10.** Trend models for Podlaskie province.

| Data/Model | Annual Data | Monthly Data |
|---|---|---|
| Exponential | $y = 1605.6e^{-0.057x}$ | $y = 1308.3e^{-4E-06x}$ |
| | $R^2 = 0.9468$ | $R^2 = 0.0019$ |
| Linear | $y = -49.516x + 1456.8$ | $y = -0.0061x + 1364.2$ |
| | $R^2 = 0.9544$ | $R^2 = 0.004$ |
| Logarithmic | $y = -359.1\ln(x) + 1688.1$ | $y = -260.2\ln(x) + 3878.5$ |
| | $R^2 = 0.8715$ | $R^2 = 0.0042$ |
| Polynomial of 2nd degree | $y = 0.8747x^2 - 68.759x + 1530.5$ | $y = 6E-06x^2 - 0.5033x + 11742$ |
| | $R^2 = 0.963$ | $R^2 = 0.0116$ |
| Polynomial of 3rd degree | $y = -0.0076x^3 + 1.1243x^2 - 71.007x + 1535.1$ | $y = 3E-09x^3 - 0.0003x^2 + 12.994x - 176022$ |
| | $R^2 = 0.9631$ | $R^2 = 0.0144$ |
| Polynomial of 4th degree | $y = -0.0326x^4 + 1.4274x^3 - 19.501x^2 + 35.49x + 1393.7$ | $y = -1E-11x^4 + 2E-06x^3 - 0.1254x^2 + 3495.8x - 4E+07$ |
| | $R^2 = 0.9721$ | $R^2 = 0.1273$ |
| Polynomial of 5th degree | $y = -0.0041x^5 + 0.1948x^4 - 3.0764x^3 + 19.076x^2 - 97.781x + 1523.2$ | $y = 3E-15x^5 - 6E-10x^4 + 5E-05x^3 - 2.1244x^2 + 45221x - 4E+08$ |
| | $R^2 = 0.9759$ | $R^2 = 0.1385$ |
| Polynomial of 6th degree | $y = 0.0003x^6 - 0.0258x^5 + 0.7421x^4 - 9.6906x^3 + 57.881x^2 - 196.26x + 1598.8$ | $y = 6E-18x^6 - 2E-12x^5 + 2E-07x^4 - 0.0091x^3 + 282.88x^2 - 5E+06x + 3E+10$ |
| | $R^2 = 0.9765$ | $R^2 = 0.2485$ |
| Potentiometric | $y = 2004.2x^{-0.392}$ | $y = 7319.3x^{-0.178}$ |
| | $R^2 = 0.7814$ | $R^2 = 0.002$ |

**Table A11.** Trend models for Pomeranian province.

| Data/Model | Annual Data | Monthly Data |
|---|---|---|
| Exponential | $y = 3663.6e^{-0.026x}$ | $y = 84.959e^{8E-05x}$ |
| | $R^2 = 0.7854$ | $R^2 = 0.417$ |
| Linear | $y = -69.164x + 3559.7$ | $y = 0.1573x - 4551.4$ |
| | $R^2 = 0.8237$ | $R^2 = 0.4341$ |
| Logarithmic | $y = -496.7\ln(x) + 3872.3$ | $y = 6541.5\ln(x) - 67579$ |
| | $R^2 = 0.7377$ | $R^2 = 0.4306$ |
| Polynomial of 2nd degree | $y = -1.0377x^2 - 46.334x + 3472.2$ | $y = 2E-05x^2 - 1.8844x + 38064$ |
| | $R^2 = 0.8291$ | $R^2 = 0.455$ |
| Polynomial of 3rd degree | $y = -0.8089x^3 + 25.655x^2 - 286.73x + 3963.3$ | $y = -3E-08x^3 + 0.0033x^2 - 139.08x + 2E+06$ |
| | $R^2 = 0.9203$ | $R^2 = 0.5015$ |
| Polynomial of 4th degree | $y = -0.0349x^4 + 0.7259x^3 + 3.5961x^2 - 172.83x + 3812.1$ | $y = -2E-11x^4 + 3E-06x^3 - 0.1737x^2 + 4788.7x - 5E+07$ |
| | $R^2 = 0.9249$ | $R^2 = 0.5384$ |
| Polynomial of 5th degree | $y = -0.002x^5 + 0.0738x^4 - 1.4261x^3 + 22.029x^2 - 236.51x + 3873.9$ | $y = 1E-14x^5 - 2E-09x^4 + 0.0002x^3 - 7.3691x^2 + 154981x - 1E+09$ |
| | $R^2 = 0.9253$ | $R^2 = 0.5622$ |
| Polynomial of 6th degree | $y = 1E-04x^6 - 0.0085x^5 + 0.2378x^4 - 3.4082x^3 + 33.657x^2 - 266.02x + 3896.6$ | $y = 8E-18x^6 - 2E-12x^5 + 2E-07x^4 - 0.0108x^3 + 336.03x^2 - 6E+06x + 4E+10$ |
| | $R^2 = 0.9253$ | $R^2 = 0.5882$ |
| Potentiometric | $y = 4053x^{-0.178}$ | $y = 6E-12x^{3.1379}$ |
| | $R^2 = 0.6494$ | $R^2 = 0.4138$ |

**Table A12.** Trend models for the Silesian province.

| Data/Model | Annual Data | Monthly Data |
|---|---|---|
| Exponential | $y = 8278.3e^{-0.053x}$ | $y = 1795.9e^{2E-05x}$ |
| | $R^2 = 0.9134$ | $R^2 = 0.0739$ |
| Linear | $y = -234.11x + 7432$ | $y = 0.1003x + 182.21$ |
| | $R^2 = 0.9536$ | $R^2 = 0.0874$ |
| Logarithmic | $y = -1535\ln(x) + 8173.7$ | $y = 4135.4\ln(x) - 39626$ |
| | $R^2 = 0.7117$ | $R^2 = 0.0852$ |
| Polynomial of 2nd degree | $y = -6.7177x^2 - 86.325x + 6865.4$ | $y = 5E-05x^2 - 4.2277x + 90523$ |
| | $R^2 = 0.9765$ | $R^2 = 0.134$ |
| Polynomial of 3rd degree | $y = 0.6756x^3 - 29.012x^2 + 114.46x + 6455.2$ | $y = -2E-08x^3 + 0.0029x^2 - 124.35x + 2E+06$ |
| | $R^2 = 0.9829$ | $R^2 = 0.1517$ |
| Polynomial of 4th degree | $y = -0.1059x^4 + 5.337x^3 - 96.012x^2 + 460.4x + 5995.8$ | $y = -4E-11x^4 + 6E-06x^3 - 0.3704x^2 + 10268x - 1E+08$ |
| | $R^2 = 0.9872$ | $R^2 = 0.2328$ |
| Polynomial of 5th degree | $y = -0.0031x^5 + 0.0633x^4 + 1.9849x^3 - 67.299x^2 + 361.21x + 6092.1$ | $y = 2E-14x^5 - 4E-09x^4 + 0.0003x^3 - 13.992x^2 + 294603x - 2E+09$ |
| | $R^2 = 0.9873$ | $R^2 = 0.275$ |
| Polynomial of 6th degree | $y = 0.0019x^6 - 0.1298x^5 + 3.2675x^4 - 36.739x^3 + 159.89x^2 - 215.37x + 6535.2$ | $y = 2E-17x^6 - 5E-12x^5 + 5E-07x^4 - 0.0288x^3 + 897.12x^2 - 1E+07x + 1E+11$ |
| | $R^2 = 0.9882$ | $R^2 = 0.3657$ |
| Potentiometric | $y = 9554.3x^{-0.337}$ | $y = 0.4145x^{0.87}$ |
| | $R^2 = 0.6368$ | $R^2 = 0.0719$ |

**Table A13.** Trend models for the Holy Cross province.

| Data/Model | Annual Data | Monthly Data |
|---|---|---|
| Exponential | $y = 2545.8e^{-0.047x}$ | $y = 1906e^{-1E-05x}$ |
| | $R^2 = 0.9269$ | $R^2 = 0.0241$ |
| Linear | $y = -70.049x + 2347.9$ | $y = -0.0142x + 1713.4$ |
| | $R^2 = 0.9651$ | $R^2 = 0.0243$ |
| Logarithmic | $y = -489.1\ln(x) + 2634.3$ | $y = -610.5\ln(x) + 7615.9$ |
| | $R^2 = 0.8169$ | $R^2 = 0.0258$ |
| Polynomial of 2nd degree | $y = -0.2957x^2 - 63.544x + 2323$ | $y = 2E-05x^2 - 1.5293x + 33338$ |
| | $R^2 = 0.9656$ | $R^2 = 0.1037$ |
| Polynomial of 3rd degree | $y = -0.0844x^3 + 2.4883x^2 - 88.617x + 2374.2$ | $y = -4E-10x^3 + 7E-05x^2 - 3.8804x + 66046$ |
| | $R^2 = 0.9667$ | $R^2 = 0.1037$ |
| Polynomial of 4th degree | $y = -0.0538x^4 + 2.2813x^3 - 31.514x^2 + 86.947x + 2141$ | $y = -1E-11x^4 + 2E-06x^3 - 0.1485x^2 + 4133.1x - 4E+07$ |
| | $R^2 = 0.9791$ | $R^2 = 0.2824$ |
| Polynomial of 5th degree | $y = -0.0025x^5 + 0.083x^4 - 0.4272x^3 - 8.3145x^2 + 6.8028x + 2218.9$ | $y = 4E-15x^5 - 8E-10x^4 + 7E-05x^3 - 2.8272x^2 + 60046x - 5E+08$ |
| | $R^2 = 0.9798$ | $R^2 = 0.3051$ |
| Polynomial of 6th degree | $y = 0.0006x^6 - 0.0449x^5 + 1.1552x^4 - 13.385x^3 + 67.706x^2 - 186.13x + 2367.2$ | $y = 5E-18x^6 - 1E-12x^5 + 1E-07x^4 - 0.0079x^3 + 245.49x^2 - 4E+06x + 3E+10$ |
| | $R^2 = 0.981$ | $R^2 = 0.3988$ |
| Potentiometric | $y = 2996.8x^{-0.316}$ | $y = 412018x^{-0.556}$ |
| | $R^2 = 0.7194$ | $R^2 = 0.0257$ |

**Table A14.** Trend models for the Warmian–Masurian province.

| Data/Model | Annual Data | Monthly Data |
|---|---|---|
| Exponential | $y = 2216.7e^{-0.029x}$ | $y = 3353.1e^{-2E-05x}$ |
| | $R^2 = 0.6897$ | $R^2 = 0.0613$ |
| Linear | $y = -43.531x + 2119.9$ | $y = -0.0281x + 2606.2$ |
| | $R^2 = 0.699$ | $R^2 = 0.0597$ |
| Logarithmic | $y = -256.3\ln(x) + 2194.8$ | $y = -1178\ln(x) + 13969$ |
| | $R^2 = 0.4206$ | $R^2 = 0.0602$ |
| Polynomial of 2nd degree | $y = -3.7986x^2 + 40.037x + 1799.5$ | $y = 5E\text{-}06x^2 - 0.4703x + 11837$ |
| | $R^2 = 0.8541$ | $R^2 = 0.0639$ |
| Polynomial of 3rd degree | $y = -0.0283x^3 - 2.8652x^2 + 31.631x + 1816.7$ | $y = -1E\text{-}09x^3 + 0.0002x^2 - 7.8062x + 113891$ |
| | $R^2 = 0.8543$ | $R^2 = 0.0645$ |
| Polynomial of 4th degree | $y = -0.033x^4 + 1.4255x^3 - 23.761x^2 + 139.52x + 1673.4$ | $y = -1E\text{-}11x^4 + 2E\text{-}06x^3 - 0.1233x^2 + 3430.8x - 4E+07$ |
| | $R^2 = 0.8631$ | $R^2 = 0.1418$ |
| Polynomial of 5th degree | $y = -0.0037x^5 + 0.1709x^4 - 2.6132x^3 + 10.833x^2 + 20.015x + 1789.5$ | $y = 7E\text{-}15x^5 - 1E\text{-}09x^4 + 0.0001x^3 - 5.2879x^2 + 111231x - 9E+08$ |
| | $R^2 = 0.866$ | $R^2 = 0.1945$ |
| Polynomial of 6th degree | $y = 0.0013x^6 - 0.0879x^5 + 2.2988x^4 - 28.33x^3 + 161.71x^2 - 362.89x + 2083.8$ | $y = 7E\text{-}18x^6 - 2E\text{-}12x^5 + 2E\text{-}07x^4 - 0.0095x^3 + 297.52x^2 - 5E+06x + 3E+10$ |
| | $R^2 = 0.8748$ | $R^2 = 0.2818$ |
| Potentiometric | $y = 2322.2x^{-0.17}$ | $y = 1E+07x^{-0.862}$ |
| | $R^2 = 0.4065$ | $R^2 = 0.0618$ |

**Table A15.** Trend models for Greater Poland province.

| Data/Model | Annual Data | Monthly Data |
|---|---|---|
| Exponential | $y = 5383.4e^{-0.039x}$ | $y = 2269.7e^{6E-06x}$ |
| | $R^2 = 0.6153$ | $R^2 = 0.0067$ |
| Linear | $y = -146.33x + 5263.9$ | $y = 0.0232x + 1999.8$ |
| | $R^2 = 0.6566$ | $R^2 = 0.0109$ |
| Logarithmic | $y = -1137\ln(x) + 6111$ | $y = 924.51\ln(x) - 6869.2$ |
| | $R^2 = 0.6881$ | $R^2 = 0.0099$ |
| Polynomial of 2nd degree | $y = 10.982x^2 - 387.94x + 6190$ | $y = 4E\text{-}05x^2 - 3.4456x + 74403$ |
| | $R^2 = 0.7643$ | $R^2 = 0.0807$ |
| Polynomial of 3rd degree | $y = 1.0563x^3 - 23.874x^2 - 74.018x + 5548.7$ | $y = -2E\text{-}08x^3 + 0.0031x^2 - 129.52x + 2E+06$ |
| | $R^2 = 0.792$ | $R^2 = 0.126$ |
| Polynomial of 4th degree | $y = -0.247x^4 + 11.924x^3 - 180.07x^2 + 732.49x + 4477.5$ | $y = -2E\text{-}11x^4 + 4E\text{-}06x^3 - 0.2567x^2 + 7101.8x - 7E+07$ |
| | $R^2 = 0.8328$ | $R^2 = 0.2176$ |
| Polynomial of 5th degree | $y = -0.0718x^5 + 3.7046x^4 - 66.349x^3 + 490.37x^2 - 1583.6x + 6728$ | $y = 8E\text{-}15x^5 - 2E\text{-}09x^4 + 0.0001x^3 - 6.2467x^2 + 132133x - 1E+09$ |
| | $R^2 = 0.9236$ | $R^2 = 0.2366$ |
| Polynomial of 6th degree | $y = -0.0021x^6 + 0.0687x^5 + 0.1521x^4 - 23.418x^3 + 238.5x^2 - 944.4x + 6236.8$ | $y = 2E\text{-}17x^6 - 4E\text{-}12x^5 + 4E\text{-}07x^4 - 0.0231x^3 + 723.06x^2 - 1E+07x + 8E+10$ |
| | $R^2 = 0.9256$ | $R^2 = 0.3722$ |
| Potentiometric | $y = 6695.8x^{-0.301}$ | $y = 215.04x^{0.246}$ |
| | $R^2 = 0.6259$ | $R^2 = 0.006$ |

**Table A16.** Trend models for the West Pomeranian province.

| Data/Model | Annual Data | Monthly Data |
|---|---|---|
| Exponential | $y = 2381.2e^{-0.04x}$ | $y = 389.87e^{3E-05x}$ |
| | $R^2 = 0.939$ | $R^2 = 0.0946$ |
| Linear | $y = -60.835x + 2250.1$ | $y = 0.0534x - 685.14$ |
| | $R^2 = 0.9558$ | $R^2 = 0.1096$ |
| Logarithmic | $y = -430\ln(x) + 2510.2$ | $y = 2186.6\ln(x) - 21717$ |
| | $R^2 = 0.8291$ | $R^2 = 0.1053$ |
| Polynomial of 2nd degree | $y = 0.022x^2 - 61.319x + 2252$ | $y = 4E\text{-}05x^2 - 3.6238x + 76071$ |
| | $R^2 = 0.9558$ | $R^2 = 0.2583$ |
| Polynomial of 3rd degree | $y = -0.0448x^3 + 1.4998x^2 - 74.628x + 2279.2$ | $y = -2E\text{-}08x^3 + 0.0029x^2 - 121.55x + 2E{+}06$ |
| | $R^2 = 0.9562$ | $R^2 = 0.3334$ |
| Polynomial of 4th degree | $y = -0.0181x^4 + 0.7514x^3 - 9.9445x^2 - 15.538x + 2200.7$ | $y = -1E\text{-}11x^4 + 2E\text{-}06x^3 - 0.1535x^2 + 4230.3x - 4E{+}07$ |
| | $R^2 = 0.9581$ | $R^2 = 0.3963$ |
| Polynomial of 5th degree | $y = -0.0034x^5 + 0.1701x^4 - 2.9764x^3 + 21.986x^2 - 125.85x + 2307.9$ | $y = 9E\text{-}15x^5 - 2E\text{-}09x^4 + 0.0002x^3 - 6.3676x^2 + 133940x - 1E{+}09$ |
| | $R^2 = 0.9598$ | $R^2 = 0.4351$ |
| Polynomial of 6th degree | $y = 0.0002x^6 - 0.0185x^5 + 0.5508x^4 - 7.5768x^3 + 48.976x^2 - 194.34x + 2360.5$ | $y = 5E\text{-}18x^6 - 1E\text{-}12x^5 + 1E\text{-}07x^4 - 0.0066x^3 + 205.46x^2 - 3E{+}06x + 2E{+}10$ |
| | $R^2 = 0.96$ | $R^2 = 0.4567$ |
| Potentiometric | $y = 2763.2x^{-0.272}$ | $y = 0.001x^{1.3345}$ |
| | $R^2 = 0.7568$ | $R^2 = 0.0905$ |

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
