# Peer review of "Forecasting the Number of Road Accidents in Polish Provinces Using Trend Models"

_applsci, doi:10.3390/app13052898_

Round 1

Reviewer 1 Report

find the attached file

Author Response

Dear Sir, Thank you very much for all your valuable comments.
Taking into account your comments, I have made the following changes in the article:
1. I have reorganized the entire article.
2. corrected the abstract
3) I added citations in the paper.
4. I modified the literature review and added interesting papers
5. I renumbered the points.
6. I modified tables 1.
7. I added scales and north to figure 2.
8. in the tables from R2, I changed the comma to a period
9. I modified the summary.
10. modified the sources in terms of mdpi
In the article I analyzed, I used the trend method to show how we can use the method in question when forecasting the number of traffic accidents.  While other forecasting methods such as exponential smoothing and neural networks. He is currently preparing a monograph on the subject, comparing the use of different forecasting methods. 
I have highlighted changes in yellow.

Reviewer 2 Report

No novelty in the paper. Plenty of tables in the manuscript. No proper explanation about the work. The article should be rejected.

Author Response

Dear Sir, Thank you very much for the information. 

Reviewer 3 Report

The author presented the manuscript Forecasting the number of road accidents in Poland province using trend models aimed at assessing and predicting the frequency of road accidents with accidents throughout Poland. The work processed a significant array of data for a 20-year period for 16 subjects. The author gives an interesting review of works on similar topics, in which he reveals the advantages and disadvantages of statistical data processing methods. Interesting results of the work are:

  - The number of traffic accidents has been decreasing in recent years.

- The used trend model is suitable for fast forecasting of annual averages, with a fairly high level of accuracy.

In order to improve the manuscript, we offer the authors:

1. Correct the table 1. The table is cumbersome and poorly displays the data (we suggest thinking about a different way of presenting the material you publish).

2. Table 1 and table 2 partially duplicate each other. Please correct.

3. Figure 1 carries only information about the geographical location of subjects within the country. Perhaps by combining with the data from table 2 you will get a better result.

4. On line 122 you say that the number of accidents in Poland is higher than in the European Union, but you do not provide data/values/references.

5. Give an explanation of what the value in figure 2 is measured in (add a unit of measurement, and also make it clearer.). Perhaps it makes sense to put the highest values on the drawing, or sign the names of the subjects.

6. Having examined figure 4, one gets the impression that the largest number of cases occurs not in summer, but in spring. (lines 143-144). Please clarify.

In general, the manuscript, despite the indicated shortcomings and a certain brevity in presentation, is written in good scientific language. We recommend publishing manuscript after correction.

Author Response

Dear Sir, Thank you very much for all your valuable comments.
Taking into account your comments, I have made the following changes in the article:
1. I replaced Table 1 with Chart 1.
2. I added sources in line 122.
3. changed the description in Figure 3
4. the actual number of accidents is highest in summer. On the graph (with monthly data) this is not clearly visible.

I have highlighted the changes in yellow.

Reviewer 4 Report

Although the topic under study is important, it does not have any special innovation. Similar and much stronger studies have been done. The present study is very simple and does not have an advanced analysis. The author has fitted the data of accidents of different years by using various regressions. Also, the assumptions of using regression (such as independence of variables, normality of variables, independence of errors, Durbin-Watson  test, etc.) have not been examined. The software used is not a strong statistical software either.This paper is not suitable for publication.

Author Response

Dear Mr. Professors,

Thank you for your information and valuable comments. I agree that the topic is important for this reason I took up the subject. It is not innovative, however, it is quick to determine forecasts without the use of scientific, specialized software in this area. In most cases, such as Statistika software, the software is very expensive and cannot be afforded by the small research centers it represents.

Similar studies have been conducted using neural networks and exponential models as evidenced in my publications:

https://komunikacie.uniza.sk/getrevsrc.php?identification=public&mag=csl&raid=1899&type=fin&ver=4

https://www.degruyter.com/document/doi/10.1515/eng-2022-0370/html

In addition, since the study used only historical data on the number of traffic accidents, and did not consider other factors, there is a relationship between the data used, and for this reason the method discussed can be used.

Best regards

Round 2

Reviewer 1 Report

The author have made some changes. But still I am not convinced by the applied conventional methods for such Sci journals. In my views it's too simple method and there is still lack of details explanations of each forecasted figures. It seems that According to previous trends the accidents going to decrease. 

Its suggested to clearly explain the contributions by mentioning the previous gaps in global research. Methods and Results need to be revised. otherwise it's not suitable for such journal publication!! 

Thanks

Author Response

Dear Mr. Professors,
Thank you for the information, 
I agree with you that the method is simple, but its advantage is the speed of determining the predicted value without the need for specialized software. In my next article I plan to refer to this article and compare it with my other articles on the subject. It is possible to use the regression method to forecast the number of traffic accidents because there is a cause-and-effect relationship between accidents at a given time. In addition, a research gap has been added. 
Greetings

Reviewer 2 Report

Done as suggested.

Author Response

Dear Mr. Professors,
Thank you for your information and help in this regard.
Greetings

Round 3

Reviewer 1 Report

 I did not find any new amendment from author side. The author still have not applied any new method and he is justifying his present method. But I could recommend that show the forecasting equations and R2 over the chart to clearly show the Chart and their equations. It will improve the readability of manuscript.

Or move some unnecessary tables to annexure to improve the interest of reader. 

Improve the conclusion by keeping in mind the direct objectives of the stusy, data, applied method, main findings,  limitations and recommendations. 

Author Response

Dear Professor,
Thank you for your valuable comments. According to your suggestion in the article, I created additional tables number 2 and 3 with the best trend values and moved the other tables to the appendix.
Regards
Piotr Gorzelańczyk
